# Stability and Change in Longitudinal Associations between Child Behavior Problems and Maternal Stress in Families with Preterm Born Children, Follow-Up after a RCT-Study

**DOI:** 10.3390/children6020019

**Published:** 2019-01-31

**Authors:** Inger P. Landsem, Bjørn H. Handegård, Per I. Kaaresen, Jorunn Tunby, Stein E. Ulvund, John A. Rønning

**Affiliations:** 1Child- and Adolescent Department, University Hospital of North Norway, 9016 Tromsø, Norway; per.ivar.kaaresen@unn.no; 2Department for Health Scienses, UIT the Arctic University of Norway, 9019 Tromsø, Norway; bjorn.helge.handegaard@uit.no (B.H.H.); Jorunn.tunby@uit.no (J.T.); s.e.ulvund@iped.uio.no (S.E.U.); john.ronning@uit.no (J.A.R.); 3Department of Education, Institute of pedagogics, University of Oslo, 0317 Oslo, Norway

**Keywords:** prematurity, parent–child interaction, parenting stress, behavior problems, early intervention, longitudinal

## Abstract

The Tromsø Intervention Study on Preterms (TISP) randomized 146 preterm-born children either to the Mother-Infant Transaction Program (MITP) or to a preterm control group. Previously, significant reductions of child behavior problems and maternal stress have been reported in the intervention group. This follow-up study examines whether the MITP may have affected the longitudinal adaptation between mothers and their children from two until nine years, expressed as associations between different behavioral problems and parenting stress reported by mothers. Associations between internalizing, attentional, and social problems and different dimensions of parenting stress were analyzed in separate models that included effects of time and group status. The MITP did not influence the development of longitudinal associations as no significant three-way interaction (stress*group*time) was found. Significant stress by group interactions was only found in reports on children’s attentional problems when analyzed with parent- or interaction-related stress. Mothers who had participated in the MITP reported weaker stress–behavior associations than control mothers. This effect was moderated by two independent variables, namely children’s birthweight and years of maternal education for the parent–child difficult interaction stress.

## 1. Introduction

Many interventions, aimed at improving developmental outcomes for children born prematurely have been designed and trialed in recent decades [1,2]. The Mother-Infant Transaction Program (MITP) was an early one, which has also documented long-lasting effects on child and parent wellbeing after participation [3,4,5,6,7]. The present study investigates how a modified version of the MITP (MITP-m) may affect the reciprocal adaptation between the child and his/her parents, by focusing on associations between parenting stress and child behavior problems as reported by mothers of preterm born children from two to nine years of age. 

Parenting stress is an important measure in studies investigating children’s caregiving environment, and is frequently elevated in families of prematurely born children (hereafter named preterms), especially in toddlerhood [8,9]. While parental responsiveness and caring competences are key factors in establishing a good caring environment, high levels of parenting stress are known to affect parenting behavior negatively [10]. In addition, parenting stress is described as having independent, negative effects on the parent–child relationship and child wellbeing because it negatively affects parents’ appraisal of their children’s behavior [11]. Parenting stress and child behavior problems are thus important indicators that provide information about the continuous, ongoing adaptation in parent–child relationships [8,12,13,14]. 

Preterms’ neurobiological immaturity at birth and the elevated levels of distress they experience in the perinatal period influence their behaviors [15]. Compared to term-borns, they frequently express needs of regulative support, time-outs, and parental attention in more subtle ways [16]. Examples may be frequent state changes, they tire more easily and may seem unavailable for social interactions because they quickly become fuzzy or drowsy as infants [17,18]. Later in toddlerhood, they are frequently reported as less able to establish sleep and feeding regularity, and less able to show sustained attention compared with term-borns at similar ages [19]. Poor regularity is challenging for parents. It increases the risk of parental misunderstandings and breakdowns in parent–child interactions [20,21]. Accordingly, preterms are affected by multiple additional risks, e.g., higher incidence of neonatal morbidity, more regulation difficulties, greater difficulty in parental interpretation of behavior, and a greater negative influence of dysfunctional parenting and/or interactional mismatches [22,23]. This mix of risk factors is associated with children’s maturation and adaptive behavior across childhood, and highlights why preterms frequently are considered as immature even at growing ages, with behavior sometimes observed and rated as problematic [24,25]. 

Deater-Deckard and Bulkley [1] described improved maternal confidence and satisfaction as two major optimizing effects of the MITP. On the other hand, Gerstein & Poehlmann-Tynan described key elements of the MITP as first helping parents to process their grief and difficult experiences and then enabling parents to appreciate the unique characteristics of their child and become sensitive to infant cues and readiness for interaction [22]. Recently, the MITP intervention has been described as one of few promising interventions for preterms and their families in a meta-review of systematic reviews [7]. Thus, there is reason to believe that the MITP increased parents’ understanding of why poorly adaptive behavior occurred, and thus helped them to find adaptive, coping strategies that fit their child [8]. Accordingly, parents in the preterm control group, who consistently reported more competence-related parenting stress [26], may have had fewer and less suitable coping strategies available and were thus more negatively affected by their child’s behavior [10,27]. 

Three aspects of behavioral problems, closely related to regulation difficulties, are repeatedly reported as more frequent in samples of preterms compared with term-born peers, namely internalizing, social and attentional problems [28]. Thus, these aspects of child behavior seem especially relevant to investigate in relation to parenting stress in this study. Mothers in the preterm intervention and the preterm control group reported internalizing problems at similar levels from two until nine years [6]. On the other hand, preterm control mothers reported significantly more social and attentional problems from the age of five [29].

Less parenting stress, and to some degree fewer behavior problems reported by mothers that participated in the MITP program, indicate improved bidirectional parent–child adaptation from infancy to early school age [14]. Parenting stress is a multidimensional construct, with three dimensions tapping stress related to the child, the parent, and the parent–child relation [12]. Better maternal adaptation to children’s varying levels of behavior problems after participation in the MITP might affect the associations between child behavior and the dimensions of maternal stress differently within families, depending on their initial level of stress.

This study explores the longitudinal relations between maternal ratings of child behavior problems and their concurrent reports of parenting stress. Weaker covariation is hypothesized between maternal stress and child behavior problems among mothers in the intervention group than in the control group, because these mothers have more knowledge and therefore respond more adequately when their child expresses adaptive difficulties. Thus, this study questions whether there were differences between the two preterm groups in how the association between stress and behavior developed over time, from age two to nine, and to what degree possible differences were moderated by child or maternal factors such as birthweight, gender, or maternal education.

## 2. Material and Methods

Preterms (birthweight (BW) < 2000 grams) born between March 1999 and September 2002 were randomized to either the MITP intervention or the usual follow-up at a university hospital in northern Norway, after written consent from their parents [4]. Participants were recruited while they were patients in a neonatal intensive care unit (NICU), approximately six to eight weeks before term. Children without congenital anomalies (e.g., Down syndrome), whose mothers spoke Norwegian, and who were not born as triplets were eligible. The initial study sample consisted of 1469 preterms, randomized into two groups: an intervention group (PI group, *n* = 72) and a preterm control group (PC group, *n* = 74). The PI group participated in the MITP program, starting approximately one week before estimated discharge from the NICU for each family. In addition, all families in the PI and the PC group followed the NICU guidelines for discharge of preterm infants. Depending on the degree of prematurity, this consisted of tests of visual and motor functioning, recommendations about nutrition and information about common challenges parents meet when arriving home with a preterm baby. In addition, all parents participated in a baby massage session led by a child physiotherapist. Table 1 presents birth, medical and demographic information for each study group. Randomization resulted in well-balanced groups with one exception. Mothers in the PI group had significantly more education than PC mothers (mean difference 1.1 years). 

The intervention ended three months after discharge. All participants received the same developmental and psychosocial assessments at corrected ages of 6 months and 1, 2, 3, 5, 7, and 9 years, and medical assessments until age two as part of the longitudinal follow-up. Participants were free to withdraw from further participation at any time. Withdrawal rates were low: 88% of preterm children were still participating at nine years (PI group = 67 and PC group = 62). The regional committee for medical ethics and the Norwegian Data Protection Authority (1999, 2005, and 2010) approved the study several times. The registration number in ClinicalTrials.gov is NCT00222456.

### 2.1. Design of the Intervention

Rauh, Nurcombe, Achenbach & Howell outlined the original MITP [30]. This parent guiding program consists of 12 one-hour sessions where parents and the newborn child meet with an intervention guide. One of eight specially trained nurses conducted all sessions for each family. The first seven sessions before discharge from hospital had different agendas. Parents actively participated in investigations and demonstrations of the infant’s social competencies, the infant’s signs of stress and stability in the homeostatic system, the motor system, the stability of states and the infant’s ability of alertness, responsiveness, and self-regulation. Across the following four home visits—at 1, 2, 4, and 12 weeks post-discharge—all topics were repeated and nuanced [30]. A translated and carefully adapted version of the MITP was implemented in this study, where two elements differed from the original MITP version. An initial session was included where parents met their interventionist and could air their thoughts and feelings related to the birth of the child [4]. In contrast to the Vermont study, no families received logbooks describing the sessions at the end of the program. All interventionists wrote logs that were reviewed by the study director to ensure consistent implementation of the program. All PI mothers participated in all MITP-m sessions. PI fathers participated less, on average in six of the twelve sessions. Reports from fathers are not included in this paper.

### 2.2. Data Collection

Child behavioral problems were assessed with maternal reports on the widely used Child Behavioral Checklist (CBCL/2-3 and CBCL/4-18) at the corrected ages of 2, 3, 5, 7, and 9 years [24,25]. The CBCL/2-3 questionnaire lists 99 questions, whereas the CBCL/4-18 lists 113 questions. In both questionnaires, most questions were loaded on two main dimensions (internalizing and externalizing behavior). This study used summarized CBCL T-scores of internalizing problems (which include the subdimensions addressing withdrawn, anxious, and depressed behavior) and two separate dimensions in the CBCL addressing attentional and social problems. The use of T-scores made it possible to merge data from both versions of the CBCL questionnaires in the analysis of internalizing and attentional problems. Internalizing and attentional problems were reported at all follow-ups from two until nine years, while social problems are a defined subdimension in CBCL/4-18, and was thus reported from five to nine years. 

Information about maternal stress was assessed with the Parenting Stress Index (PSI) at the children’s ages of 2, 3, 5, and 7 years and with the Parenting Stress Index-Short Form (PSI-sf) at age nine [12,31]. The PSI-sf consists of 36 questions extracted from the PSI and correlation between total stress scores on these two measures is reported to be high (0.87) (ibid). Parenting stress from all follow-ups could thus be included in the longitudinal analysis by extracting the answers to the 36 questions that are common to the PSI and PSI-sf questionnaires. Separate analyses were conducted for each of the P-C-R dimensions, Parental Distress (PD), Difficult Child (DC), and Parent–child Dysfunctional Interaction (P-CDI). The PD-dimension consists of questions about stress related to the parent’s perception of his/her parenting role, the DC-dimension of questions related to behaviors of the child, while the P-CDI-dimension includes questions about the parent–child relation and their interactions. Cronbach’s alpha for this sample on the different dimensions of PSI-SF and at different ages: for the DC-dimension between 0.78 and 0.92; for the PD-dimension between 0.84 and 0.90; and for the P-CDI-dimension between 0.78 and 0.88 at children’s ages of 2, 3, 5, 7, and 9 years. Demographic information (mother’s age, years of education, annual income, number of siblings, etc.) was reported before discharge from hospital. Neonatal and medical information was collected from the child’s medical record. 

### 2.3. Plan of Analysis

Linear mixed models (LMM) analysis is often used to analyze changes in human behavior over time, and seems particularly useful in analyses where individuals vary in both initial status and rates of change, and where repeated observations are nested within persons [32]. Data were prepared for longitudinal analysis in IBM SPSS Statistics 23, as described by Peugh and Enders [33]. Separate analyses were conducted for each behavior reported (internalizing, attentional, and social behavior problems), and with each dimension of stress (PD, DC, and P-CDI) as an independent variable. Thus, nine models were built to analyze maternal reports of stress and child behavior problems. The time variable was coded as the number of years from baseline, and baseline defined as the age when the first measures in the analysis were reported. Baseline was set at the age of two in the analyses of internalizing and attentional problems and at age five in analyses of social problems. The intercepts could thus be interpreted as the expected behavior problems at baseline if no parenting stress was reported [33].

To assess the possibility of curvilinear change in child behavior, we tested models with linear, quadratic, and cubic time variables. In this case, model fit differences between the two competing models were compared by assessing the change in –2Log Likelihood (–2LL). Level 1 (the measurement level) in the LMM modeled the child behavior as a function of time, stress, and time-by-stress interaction, and on level 2 (the individual level), the group variable was introduced as predictor of the random (and nonrandomly varying) level 1 regression coefficients. If a three-way interaction was nonsignificant, a simpler model without the second-order interaction was fitted. The random effect connected to the square of time could not be estimated, but random effects for the intercept and the time variable (the slope) were included in level 2 of the model, because children were to be assessed with different levels of problems at baseline, and to show individual patterns of linear change in problems across childhood. The level of significance was 0.05. As recommended by Singer and Willett, interpretation of significant results was supported by inspections of scatterplots [33].

### 2.4. Ethical Approval

The Norwegian committee for medical ethics, region North (2010/2153/REK Nord, date 25.08.2010), and the Norwegian Data Protection Authority (project nr. 4275, 06.11.1998; ref. 2003/816-2 RBV/-, 24.06.2003; ref. 9472 SS/RH, 12.01.2005) approved the study in 1999, 2005, and 2010. Clinical Trials gov NCT00222456.

## 3. Results

### 3.1. Initial Exploration of Correlations

The results emanate from analyses where the three P-C-R stress dimensions (child-, parent-, and relation-related stress) functioned as independent variables in separate longitudinal investigations of associations with internalizing, social, and attentional behavior problems. Table 2 displays correlations between variables included in the analysis.

In line with the research questions, we focus on whether the association between parenting stress and child behavior problems depends on time (children’s age), and whether treatment (participation in the MITP-m) affects the stress–behavior association over time as reported by mothers. In cases where there were no time*stress*group interactions but a significant group*stress interaction, new models were created to test possible moderating effects of child and maternal factors. These analyses were only included if new models significantly improved the model fit (-2LL) of the original model. Results are presented separately for each aspect of child behavior.

### 3.2. Longitudinal Associations between Children’s Internalizing Behavior and Parenting Stress

No significant interactions (time*stress*group, time*stress, group*stress, or time*group) were detected in the three analyses investigating maternal ratings of internalizing problems (Table A1, Appendix A). This indicates stability in associations between maternal stress and child internalizing behavior over time in both groups. 

### 3.3. Longitudinal Associations between Children’s Social Problems and Parenting Stress 

Group affiliation did not affect longitudinal associations between maternal ratings of social problems and maternal stress, for any of the three stress dimensions (Table A2, Appendix A). All dimensions of parenting stress were significantly associated with mothers’ ratings of children’s social problems.

### 3.4. Longitudinal Associations between Children’s Attentional Problems and Three Dimensions of Parenting Stress

There were no significant three-way interactions (overall effects of time*stress*group and time^2^*stress*group on attentional problems). This indicates that how stress and attention problems are associated over time does not depend on group affiliation in reports from mothers. Thus, the three-way interaction variables were excluded from the following analyses. 

Initial comparison of models with attentional problems as the dependent variable showed significantly better model fit after inclusion of a quadratic time variable in reports from mothers. Thus, “time” is represented by two variables in these analyses. The first analysis, focusing on the association between attentional problems and child-related stress, detected significant interactions between stress and both time variables (stress*time and stress*quadratic time) (Table 3). Thus, the age of the child affects the association between child-related stress and attentional problems, as this becomes stronger when the child’s age increases, and this conclusion holds in both groups. 

The next two analyses involved parent- and relation-related stress, respectively (Table 3, columns two and three). These detected stronger associations between maternal stress and attentional problems in reports from PC mothers than in those from PI mothers (group*stress interactions), and this comply with differences in correlations as shown in Table 2.

A comparison of regression coefficients on the stress–behavior association at each age for the two groups of mothers separately illustrates these interactions. This is exemplified for relation-related stress in Figure 1.

Associations between parent-related stress and child attentional problems were not significantly moderated by any child or maternal factors (BW, GA, gender, medical risk, years of maternal education, and marital status). On the other hand, two moderating variables created significant three-way interactions with group and stress when tested with the parent–child difficult interaction dimension (PSI-P-CDI). These were a small effect of children’s birthweight (t (580) = 1.9, *p* = 0.048) and a somewhat stronger effect of the years of maternal education variable (t (562) = –2.7, *p* = 0.007). 

## 4. Discussion

This study investigated whether the MITP-m, a short, structured parent guidance program designed for families with preterms, may influence longitudinal parent–child bidirectional adjustments, expressed as associations between maternal stress and child behavior problems from the children’s age of two until nine years. 

The MITP-m did not modify longitudinal associations between maternal stress and child behavior problems in relation to children’s internalizing and social behavior problems. Associations between maternal stress and concurrent reports of internalizing and social problems developed with similar strength in both preterm groups. On the other hand, children’s attentional problems were more strongly associated with PC mothers’ reports of parent- and interactional-related stress than those of PI mothers in toddlerhood. Nevertheless, neither of these analyses revealed differences in longitudinal associations between the groups from toddlerhood to age nine. Thus, the only finding in this study that may be related to the PI mothers’ participation in the MITP intervention was altered relations between parent- and relation-related stress and reports of children’s attentional problems. Since the time by stress by group interactions in these models were nonsignificant, the altered relations mentioned above seemed to continue from toddlerhood, as no three-way interaction appears in these reports. This paper discusses how likely such an effect of participation in the MITP-m might be. 

Attention is a basic prerequisite for social interaction and develops across childhood, nurtured by children’s experiences [23]. At the same time, when children’s behavior is characterized by an immature and rapidly changing body language, attention may be one of the most difficult areas of behavior to interpret for parents. If mothers fail to facilitate children’s immature and often brief moments of attention, it may negatively affect their own feelings of love and attachment, because of the lack of social and emotional interactions with the child [20,34]. In line with a transactional understanding of social interactions, this may facilitate a negative bidirectional development between parent and child and increase feelings of maternal stress. Tu et al. reported an association between the quality of maternal interactive behavior and more sustained attention in children of mothers who experienced low levels of parenting stress, but this association was absent for mothers who experienced higher levels of parenting stress [15]. Thus, parenting stress interferes with the quality of maternal interactive behavior, which is even more important for preterms [35,36].

The MITP highlighted observations of infants’ early attentional capacities in almost all sessions. This was an intended strategy to support parents’ awareness of “golden moments”: moments where social parent–child interactions could take place. Olafsen et al. reported improved early social communication skills among PI children compared to PC children at the age of one [37]. On the other hand, PC mothers reported more stress related to lack of own competence than PI mothers at all follow-ups from one to seven years [26]. The MITP-m may have generated a lasting change in PI mothers’ self-efficacy in their role as parents, and this may shed light on how the burden of parent- and interaction-related stress affects PC mothers more than PI mothers at children’s age of two and at later ages. Guralnick, Hammond, Neville, and Connor found that the unique kind of support that addresses specific challenges in the care of individual children with developmental delays decreased parenting stress, while different types of general support did not [38]. The MITP-m program is a good example of such specific support.

Attentional problems (inattention, impulsivity, and hyperactivity) remain a main challenge for preterm children’s development, affecting 9 to 30% of preterms [28,39]. Attentional problems seems strongly associated with genetic factors, which also predict long-term cognitive functioning [40,41]. There is growing evidence of a causal relationship between low birth weight and attentional problems [42]. This may illuminate why children’s birthweight moderates the group difference in the association between parent–child interaction-related stress (P-CDI) and attentional problems. Inspection of scatterplots confirmed that elevated levels of attentional problems were frequently associated with heightened P-CDI stress in mothers of extremely preterm children (BW < 1000 grams) in both preterm groups. The moderating effect of birthweight on the stress–behavior association seemed to occur because PI mothers reported a weaker stress–behavior association than PC mothers for children with less prematurity (BW approximately between 1200 and 2000 grams). Thus, a weaker association between attentional problems and P-CDI stress among PI mothers than PC mothers, where the children had birthweights in the upper half of this sample, may illustrate how the MITP-m supported bidirectional mother–child adaptation, which may have had preventive effects on both maternal stress and child attentional problems. On the other hand, it may also be an effect of greater variation in stress and problem scores in the PC group.

Maternal education appears as the second moderating factor for group differences in associations between child attentional problems and P-CDI. PC mothers, with twelve years of education or less reported significantly stronger associations between attentional problems and P-CDI than PI mothers with similar years of education did. Stress–behavior associations in reports from mothers with more than 12 years of education did not differ significantly between groups. This could be an effect of a restriction of range problem, but inspection of scatterplots with attentional problems and maternal stress, split between groups and at different ages separately, did not support that assumption. Similar patterns between groups were found for child- and parent-related stress, but not creating significant interactions. Previously, preterms of mothers with no formal education beyond high school have been reported with heightened attentional problems at two years of age [39], and Hall et al. report that background factors such as parental education had a strong negative effect on parental interactive behavior in their study of two-year-old toddlers [16]. Downey et al. describe lower maternal education as an indicator of lower socioeconomic status (SES); this, in line with other low-SES factors, is associated with increased risk of attentional problems [39]. In the current study, a significant difference in years of maternal education appeared, despite randomization at inclusion. PI mothers had on average 1.1 more years of education at inclusion than PC mothers. 

On the other hand, if the MITP-m enabled a better mother–child adaptation in the PI group, resulting in weaker stress–behavior associations in toddlerhood, this may be promising. Leijten, Raaijmakers, de Castro & Matthys reported that parent training programs had beneficial effects for both disadvantaged and nondisadvantaged families, but that the beneficial effects could be weaker in disadvantaged families if the burden of problems was low initially [43]. The MITP-m started before discharge in the hospital and followed families until three months postdischarge, and this may have been especially important for PI mothers with relatively low education levels. They were offered repeated meetings to ask for advice and share observations and this knowledge may have been maintained by the repeated follow-ups for participants in this study. 

### Strengths and Limitations

This study has several limitations. The dataset consists of mothers’ responses to standardized questionnaires at five follow-ups. The mothers’ motivation to answer accurately may have been affected by their participation in examinations of the child and repeated responses to the same questionnaires. On the other hand, repeated answers on standardized questionnaires are a strength, since they give comparable information from several follow-ups. Responses to questionnaires are subjective information and parents’ ratings were not compared with objective observations or interviews. On the other hand, it lies in the definition of parenting stress that it is a subjective experience and not any kind of biological measure [8].

Another limitation is the size of the dataset, making it difficult to account for moderating factors of associations between child behavior problems and maternal stress. Even though the current study is relatively small, it is unique because it is the first to examine differences in longitudinal covariation between parenting stress and child outcomes after an early intervention. Finally, the results reported in this paper must be interpreted with caution because they refer to analyses from nine different statistical models. 

High response rates at all follow-ups resulted in few missing values. The use of LMM ensured that concurrent reported behavior and stress could be included in the analysis, even though some mothers did not respond at all follow-ups. Initial computer-generated randomization of participants created two comparable groups regarding neonatal, medical and sociodemographic variables [4]. Irrespective of group, all families received the same information and cooperated with the same study coordinator and project leader across childhood. The extended follow-up program may have influenced families in both groups positively. They have been free to ask for advice at any time and this has probably diminished group differences. 

## 5. Conclusions

Previous reports from TISP documented reductions of both parenting stress and child behavior problems after participation in the MITP-m [6,26]. This study, which focuses on stress–behavior associations within each family, investigated how associations in maternal reports vary depending on child age, for different aspects of child behavior and dimensions of parenting stress. The assumption that a weaker covariation between ratings of child behavior problems and parenting stress would be found after participation in the MITP-m program was not confirmed. Neither were there group effects on longitudinal stress–behavior associations reported by mothers in relation to internalizing or social behavior problems. On the other hand, the finding of weaker associations between attentional problems and parent- and interaction-related stress in reports from preterm intervention mothers compared to the control mothers may be a positive mechanism, triggered by the MITP-m. Previous papers from this study have reported reduced mean levels of attentional problems and maternal stress after participation in the MITP-m. This study adds an insight about some altered stress–behavior associations within unique families, namely a weaker association between attentional problems and stress affecting the parent and the parent–child interaction. More longitudinal research on parent–child bidirectional adaptation after parent training programs seems necessary. New studies need to have enough power to investigate how parenting stress at low, moderate and high levels may affect children’s long-term maturation and development. 

## Figures and Tables

**Figure 1 children-06-00019-f001:**
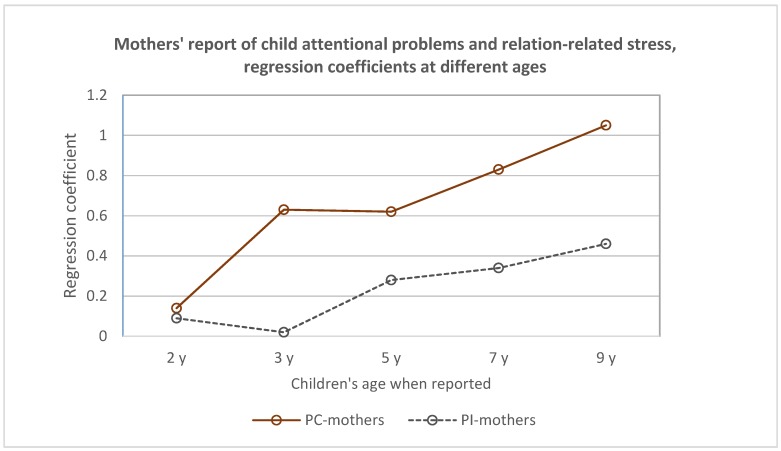
Maternal assessments of child attentional problems and their corresponding ratings of relation-related stress, regression coefficients by group. Regression coefficients computed for each group of mothers on data from each follow-up by simple regression.

**Table 1 children-06-00019-t001:** Birth, medical, and demographic information on the study sample in TISP.

	PI Group *n* = 72	PC Group *n* = 74
***Infant characteristics***		
**BW, mean ± SD, grams**	1396 ± 429	1381 ± 436
400–1000 grams, *n* (%)	20 (28)	20 (27)
1001–1500 grams, *n* (%)	15 (21)	20 (27)
1501–2000 grams, *n* (%)	37 (51)	34 (46)
**GA, mean ± SD, week**	30.2 ± 3.1	29.9 ± 3.5
<28-week, *n* (%)	17 (24)	19 (27)
28–32-week, *n* (%)	36 (50)	37 (50)
>33-week, *n* (%)	19 (26)	18 (24)
Boy, *n* (%)	38 (53)	39 (53)
Twin, *n* (%)	16 (22)	16 (21)
Received ventilation, *n* (%) Duration of ventilation, *n* (%) Postnatal steroid use, *n* (%) Oxygen therapy at 38-week GA, *n* (%)	29 (40) 7.0 ± 18.6 9 (13) 11 (15)	37 (50) 7.1 ± 17.3 10 (14) 14 (19)
**Abnormal cerebral ultrasound, *n* (%)**		
IVH grade 1 or 2	7 (10)	8 (11)
IVH grade 3 or 4	3 (4)	5 (7)
Periventricular leukomalacia	4 (6)	8 (11)
***Maternal and social characteristics***		
Mother’s age ^a^, mean ± SD	30.8 ± 6.1	29.1 ± 6.4
First-born child, *n* (%)	40 (56)	37 (54)
Mother’s education ^a^, mean ± SD, *n* = 131	14.6 ± 2.8	13.5 ± 3.2
Father’s education ^a^, mean ± SD, *n* = 131	13.8 ± 3.1	13.5 ± 3.2
Mother’s monthly income ^b^, mean ± SD, *n* = 131	15.8 ± 7.7	14.6 ± 6.7
Father’s monthly income ^b^, mean ± SD, *n* = 131	21.1 ± 8.7	19.9 ± 8.1

a = years; b = 1000 Norwegian kroner; GA = gestational age, IVH = intraventricular hemorrhage.

**Table 2 children-06-00019-t002:** Mean correlations between three aspects of child behavior and three dimensions of maternal stress reported by PI and PC mothers at 2, 3, 5, 7, and 9 years.

**Internalizing behavior**						
**Group**	**Stress dimension**	**2 y**	**3 y**	**5 y**	**7 y**	**9 y**
PC mothers	Child-related	0.44 **	0.48 **	0.54 **	0.62 **	0.71 **
	Parent-related	0.32 *	0.37 **	0.44 **	0.48 **	0.55 **
	Interaction-related	0.28	0.34 **	0.37 **	0.37 **	0.48 **
PI mothers	Child-related	0.41 **	0.37 **	0.55 **	0.41 **	0.60 **
	Parent-related	0.23	0.30 *	0.43 **	0.31 *	0.37 **
	Interaction-related	0.34 *	0.27 *	0.41 **	0.20	0.36 **
**Attentional behavior problems**						
**Group**	**Stress dimension**	**2 y**	**3 y**	**5 y**	**7 y**	**9 y**
PC mothers	Child-related	0.09	0.50 **	0.44 **	0.58 **	0.59 **
	Parent-related	0.25	0.49 **	0.23	0.45 **	0.50 **
	Interaction-related	0.17	0.54 **	0.45 **	0.48 **	0.68 **
PI mothers	Child-related	0.25	0.17	0.37 **	0.49 **	0.58 **
	Parent-related	−0.01	0.01	0.41 **	0.37 **	0.12
	Interaction-related	0.11	−0.02	0.38 **	0.35 **	0.39 **
**Social behavior problems**						
**Group**	**Stress dimension**			**5 y**	**7 y**	**9 y**
PC mothers	Child-related			0.55 **	0.52 **	0.58 **
	Parent-related			0.25	0.41 **	0.34 **
	Interaction-related			0.50 **	0.43 **	0.49 **
PI mothers	Child-related			0.57 **	0.39 **	0.69 **
	Parent-related			0.30 *	0.35 **	0.23
	Interaction-related			0.49 **	0.40 **	0.44 **

Level of significance: * = *p* < 0.05, ** = *p* < 0.01.

**Table 3 children-06-00019-t003:** Children’s attentional problems and associations with maternal parenting stress.

Fixed Effect Variables	Child-Related Stress (DC) as Independent Predictor			Parent-Related Stress (PD) as Independent Predictor			Relation-Related Stress (P-CDI) as Independent Predictor		
	Estimate	t-Value	*p*-Value	Estimate	t-Value	*p*-Value	Estimate	t-Value	*p*-Value
Intercept	**50.01**	**43.77**	**<0.0005**	**51.96**	**58.13**	**<0.0005**	**52.19**	**68.06**	**<0.0005**
Group	−0.96	−0.71	0.50	−1.54	−1.32	0.19	**−2.05**	**−1.97**	**0.04**
Time	**−1.34**	**−2.28**	**0.02**	−0.41	−0.85	0.39	−0.28	−0.71	0.49
Quadratic time	**0.22**	**3.05**	**0.003**	0.08	1.38	0.17	0.06	1.13	0.28
Group*Time	0.13	0.29	0.78	0.65	−1.45	0.15	0.36	0.78	0.42
Group* Quadratic time	0.02	0.31	0.75	−0.04	−0.70	0.49	−0.02	−0.29	0.78
Stress	**0.16**	**2.17**	**0.03**	0.02	0.28	0.78	−0.08	−0.10	0.83
Time*Stress	**0.08**	**2.32**	**0.02**	0.43	1.12	0.26	0.05	1.13	0.27
Quadratic time *Stress	**−0.01**	**−2.29**	**0.02**	-0.004	-0.91	0.36	−0.004	−0.67	0.51
Group*Stress	0.07	1.04	0.33	**0.18**	**2.29**	**0.02**	**0.34**	**3.69**	**<0.0005**

Each column reports from three separate analyses. Intercept = Predicted behavior at baseline for individuals in the PI group if stress = 0. Next: fixed effects of group, time, quadratic time and dimension of stress on attentional behavior and interactions. Bold *p*-values denote significance at *p* < 0.05 level.

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
