# Peer review of "Stability and Change in Longitudinal Associations between Child Behavior Problems and Maternal Stress in Families with Preterm Born Children, Follow-Up after a RCT-Study"

_children, 2019, doi:10.3390/children6020019_

Round 1

Reviewer 1 Report

Thank you for the opportunity to review this paper, which examines longitudinal relations between parenting stress and child behavior problems for families that were randomized to an early intervention for preterm birth or a control group. The sample unique, and the intervention piece affords an interesting opportunity to examine causal mechanisms. However, the rationale for asking the specific questions in the current study is lacking, some methodological questions must be clarified, and the results and their importance are unclear. For these reasons, I recommend against acceptance of the manuscript in its current form. More specific comments are below:

General:

1.     The use of “preterms” for prematurely born children is not person-first language. I strongly recommend not using the shorthand.

Abstract:

2.     Results are confusing as currently written. What is important and why?

Introduction:

3.     Ln. 38: “the transactional interplay in the bidirectional adaptation” is repetitive. Please rephrase for clarity.

4.     Ln. 41: caring should be caregiving

5.     Preterm should be defined at the outset – what GA is your cutoff?

6.     In general, terminology is confusing – not sure what the authors mean by “time-outs” (line 52), “fuzzy” (line 54)

7.     It’s very clear that children born preterm are at higher risk for a range of issues, but the introduction ignores that many children born preterm do just fine. And generally after infancy they are caught up with term-born peers. It is important to include this perspective as well.

8.     It’s unclear why internalizing, social, and attentional problems would be labeled as “regulation difficulties” (lines 76-77). Also in other parts of the paper these problems are broadly labeled as “behavior problems” and in the hypothesis the term used is “adaptive difficulties.” Regulation difficulties, behavior problems, and adaptive difficulties are all very different constructs, albeit all related to internalizing, social, and attentional problems but certainly not interchangeable. It is important to keep this consistent.

9.     Poor rationale overall – why is the current study an important next step given what we know about parenting stress and behavior problems

10.  It’s not clear why these 3 specific domains (internalizing, social, and attention) were selected from the range of CBCL domains – e.g., why was externalizing not examined?

11.  Bidirectional influences are discussed throughout the introduction but the study did not examine bidirectional influences

Material and methods:

12.  LARGE CRITIQUE: Why is preterm operationalized by birth weight?

13.  Provide examples of congenital abnormalities that were excluded

14.  It is not sufficient to provide a reference for the original MITP – more description is needed in the current paper. At the very least, what are the primary treatment targets?

15.  Need reliability metrics for CBCL and PSI in the current sample

16.  Description is needed for each domain of the PSI

17.  In the plan of analysis, the behavioral domains and stress domains should each be listed, (e.g., “analyses were conducted for each behavioral problem (i.e., internalizing, social, attentional))

Results:

18.  The exploration of correlations should actually be discussed in text. Also there is no indication of significance for the correlation table, so it is difficult to compare values.

Discussion:

19.  The discussion should review the results in the same order as the results section.

20.  I’m not sure what the authors mean by “behavior is frequently interspersed with incomprehensible body language” (line 268). Please rephrase for clarity.

21.  Line 283 – lack of competence on who’s part? The mother or the child?

22.  Line 291 – the Attention subscale of the CBCL does not include impulsivity and hyperactivity. Please keep the discussion specific to your finding.

23.  Again on lines 298-301 gestational age is being conflated with birth weight. They are of course correlated but they are not the same thing.

24.  It is unclear from the discussion what the most important result is, and why it is important.

Author Response

Response to Reviewer 1.

on the paper « Stability and change in longitudinal associations between child behavior problems and maternal stress in families with preterm born children», in review for publication in the journal Children.

We are very thankful for your detailed and thorough review of this paper. The response to your suggestions will be addressed point by point.

We agree that the short-name “preterms” may be viewed as inappropriate. On the other hand do we want to use this short-name in the paper to make it more easily readable.  In the second paragraph of the introduction we mention that this is used instead of “prematurely born children”  in the rest of the paper.

The abstract state that the longitudinal associations, between aspects of child behavior problems and dimensions of parenting stress, not are changed by an introduction of an early structured parent-training program. The three other reviewer’s of this paper seem satisfied with this short presentation of results, thus we have only specified that we report on children’s attentional problems in line 26.

The sentence is rephrased in the new version of this paper.

Caring is  changed to caregiving environment.

WHO define prematurity as birth before 37 completed weeks of gestation , counted from the first day in the last menstrual period. This is in line with the common understanding in this field of knowledge.

The terminology “time-out” and fuzzy is commonly used when talking about the caretaking of frail newborn infants. Time-outs is when a child can continue in an activity if he or she gets a pause. The description of a infant being fuzzy address the behavioral state when a newborn child is somewhere between alertness and crying. Both concepts are commonly used in developmental psychology but we notice that they aren’t are self-explanatory.

We agree that a lot of preterm born children succeed and display a healthy development throughout childhood. In most review’s just about two of three prematurely born children are reported with age-appropriate cognitive and socio-emotional development. Even though, a range of reports, meta- and systematic reviews conclude that populations of prematurely born children continue to have increased risks for socio- emotional and behavioral difficulties compared to term-born children at the same age. .

This point address an important discussion on how different concepts may or may not be used. In line with a transactional view of child development, we do not agree that regulation problems, behavioral problems are adaptive difficulties can’t be used in a discussion like the one presented in this paper. 

We think it is important to present these findings. The early intervention did not moderate the longitudinal patterns of bidirectional adaptation between preterm born children and their parents, but we found that the associations between attention problems and aspects of parental stress were affected.

The reason why three aspects of behaviors are investigated in this study is that these behaviors repeatedly are reported as elevated in samples of preterm born children. In our country, the occurrence of externalizing problems is not elevated in studies of preterm born children’s compared to term born children’s. This tie in with the large meta-analysis from 2009 done by Aarnoudse-Moens and colleagues.

Your views of this paper are registered.

This study was designed in 1998 and the study group selected a birthweight below 2000 gram as an inclusion criteria. Partly because this was used in the previous Vermont study and since we wanted to do a replication of that former study we used the same criteria.

One child with Down syndrome was excluded, this is now mentioned in the paper.

The MITP is now presented more clearly in section 2.1, before we present which modifications that were done in this study.

Data on reliability for PSI, as Cronbach’s Alfa are now presented in the method section 2.2 .

Each dimension of parenting stress are now shortly presented.

The revised version has included the lists you suggest.

Level of significance for each correlation is included in table 2 and commented on in the presentation of results.

The presentation of results and the review of them in the discussion section are now made congruent.

The sentence you address is rephrased in this revised version of the paper. New text, no at line 319: At the same time, when children’s behavior is characterized by an immature and rapidly changing body language, attention may be one of the most difficult areas of behavior to interpret for parents.

It is about maternal competence, the text is rephrased.

Thank you for noting about the contents included in the CBCL. In the discussion section of this paper we think it is appropriate to reflect about children’s attentional problems in a broader context. Thus, it seems adequate to mention what commonly is included in the concept “attention problems”.

And point 24. Your comments have been discussed among the authors and we chose  not to modify the text anymore as the other reviewers find it understandable..

Reviewer 2 Report

Dear Authors.

Your work is really worthy. I just have a little correction on line 266 (cancel "m").

I think that a couple of sentence on the relation between groups' differences on mother's educational level and child outcomes will be needed to better clarify your results.

Thanks for your work. 

Author Response

Response to Reviewer 2,

on the paper « Stability and change in longitudinal associations between child behavior problems and maternal stress in families with preterm born children», in review for publication in the journal Children.

We are very thankful for your review and comments of this paper.

We shortly comment on the two comments you offered:

1( the -m you wanted to be cancelled is the second part of the shortname of the intervention (MITP-m), and means that this is a modified version of the original MITP.

2) We have discussed if we should discuss the relation between group differences and years of mothers education more extensively, as you suggest. Even that may have improved the paper we have found that the discussion on line 308 to 332 already covers the most important of it.

We are very thankful for your appreciation of our work.

Reviewer 3 Report

A very good presentation of an evenly good study. Good language and text flow, clearly and

in detail presented methodology and results, relevant  up to date references . Strengths and

limitations clearly defined.

I feel that the paper could be published in its present form.

Author Response

Response to Reviewer 3,

on the paper « Stability and change in longitudinal associations between child behavior problems and maternal stress in families with preterm born children», in review for publication in the journal Children.

We are very thankful for your review and comments of this paper and thus very thankful for your appreciation of our work.

Reviewer 4 Report

Very interesting and well prepared article. A very  good presentation of an equally good study.

Good flow text, detail explanation of the methodology.

Otherwise, in my opinion the paper can be accepted inn present form.

Best wishes!

Author Response

Response to Reviewer 4,

on the paper « Stability and change in longitudinal associations between child behavior problems and maternal stress in families with preterm born children», in review for publication in the journal Children.

We are very thankful for your review of this paper and thus

very thankful for your appreciation of our work.